# ShiftAddAug: Augment Multiplication-Free Tiny Neural Network with Hybrid Computation

## Abstract

We introduce a novel training methodology termed **ShiftAddAug** aimed at enhancing the performance of multiplication-free tiny neural networks. Multiplication-free operators, such as Shift and Add, have garnered attention because of their hardware-friendly nature. They are more suitable for deployment on resource-limited platforms with reduced energy consumption and computational demands. However, multiplication-free networks usually suffer from underperformance in terms of accuracy compared to their vanilla counterpart with the same structure. ShiftAddAug uses costly multiplication to augment efficient but less powerful multiplication-free operators, improving network accuracy without any inference overhead. It puts a multiplication-free tiny NN into a large multiplicative model and encourages it to be trained as a sub-model to obtain additional supervision, rather than as an independent model. In the process of inference, only the multiplication-free tiny model is used. The effectiveness of ShiftAddAug is demonstrated through experiments in image classification, consistently resulting in significant improvements in accuracy and energy saving. Notably, it achieves up to a 4.95% accuracy improvement on the CIFAR100 compared to multiplication-free counterparts. This result far exceeds the directly trained multiplicative NNs of the same structure. Additionally, neural architecture search is used to obtain better augmentation effects and smaller but stronger multiplication-free tiny neural networks. Codes and models will be released upon acceptance.

## 1 Introduction

The application of deep neural networks (DNNs) on resource-constrained platforms is still limited due to their huge energy requirements and computational costs. However, with the increasing popularity of small computing devices such as IoT equipment(Statista, 2016), implementing DNNs directly on these devices can enhance privacy and efficiency. To obtain a small model deployed on edge devices, the commonly used techniques are pruning(Han et al., 2015b; Molchanov et al., 2017), quantization(Han et al., 2015a; Wang et al., 2019), and knowledge distillation(Hinton et al., 2015). There are also some training methods specially designed for tiny neural networks(tiny NNs). NetAug(Cai et al., 2022) believes tiny NNs tend to cause under-fitting results due to limited capacity, so it augments the tiny model by inserting it into a larger models, sharing the weights and gradients.

However, the NNs designed by the above works are all based on multiplication, which is not hardware-efficient. The common hardware design practice in computer architecture or digital signal processing tells that multiplication can be replaced by bitwise shifts and additions(Xue & Liu, 1986; Gwee et al., 2009) to achieve faster speed and lower energy consumption. Introducing this idea into NNs design, DeepShift(Elhoushi et al., 2021) and AdderNet(Chen et al., 2020) proposed ShiftConv operator and AddConv operator respectively.

This paper takes one step further along the direction of multiplication-free neural networks, proposing a method to augment tiny multiplication-free NNs by hybrid computation, which significantly improves accuracy without any inference overhead. Tiny computing devices have more severe computational, memory, and energy requirements, which makes using multiplication-free networks a good choice. However, these multiplication-free operators cannot restore all the information from

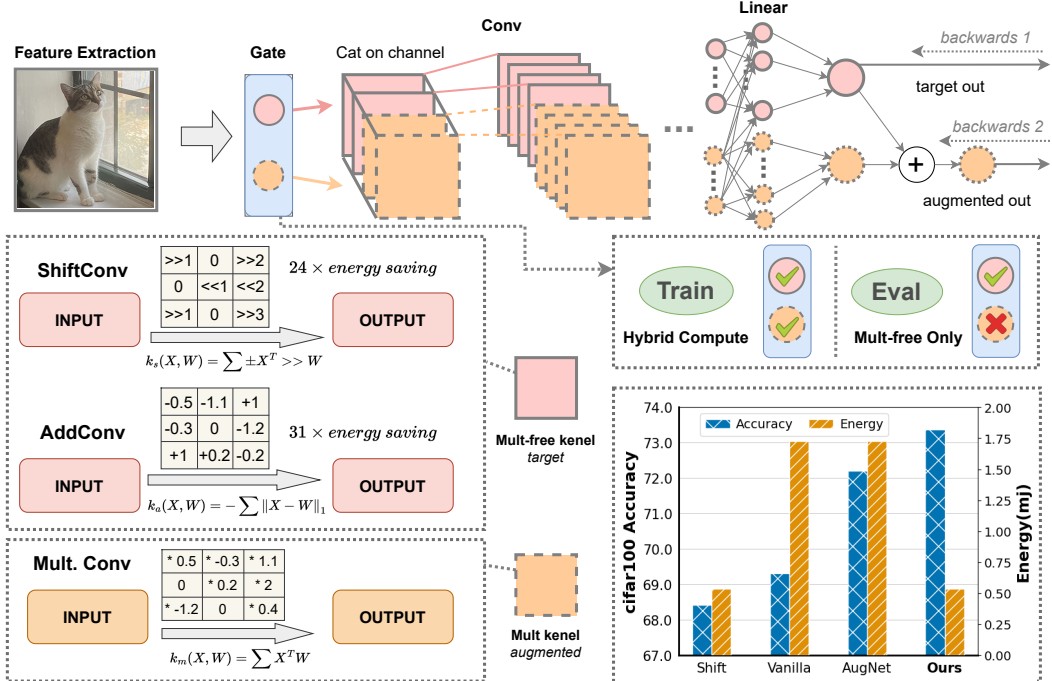

Figure 1: Overview of ShiftAddAug. The solid line and pink modules represent the multiplication-free kernels, which are used to build the target model (Shift or Add); the dotted line and orange modules represent the multiplicative kernels, which are the part that augment the model. We connect this two kinds of operators in the channel dimension for *weight sharing* and *joint training*. In the process of inference, we only retain the multiplication-free part. Obtained models have higher accuracy (up to 4.05%) than their multiplicative counterparts with the same structure, and save 68.9% energy.

the original operator, resulting in more serious under-fitting. Instead of converting and fine-tuning from a well-trained multiplicative model, we choose to build wider hybrid computing NNs, and set the multiplication-free part as the target model used in inference. We expect the stronger multiplicative part to push the target model to a better condition.

We validate our method on MobileNetV2(Sandler et al., 2018), MobileNetV3(Howard et al., 2019), MCUNet(Lin et al., 2020), ProxylessNAS(Cai et al., 2019), MobileNetV2-Tiny(Lin et al., 2020). Compared with the multiplicative networks of the same structure, we have significant accuracy improvements (1.24%~4.05%) on the CIFAR100 dataset and obtain considerable speed improvement (2.94× to 3.09×) and energy saving (67.75%~69.09%). To further improve the performance, we introduce neural architecture search(NAS) into our work, proposing a new method for searching more efficient multiplication-free tiny neural networks. Our contributions can be summarized as follows:

- For the multiplication-free tiny neural network, we propose a hybrid computing augmentation method using multiplicative operators to augment the target multiplication-free network. Under the same model structure, it is more expressive and ultra-efficient.

- We propose a new weight sharing strategy for hybrid computing augmentation, which solves the weight tearing problem in heterogeneous (e.g., Gaussian vs. Laplacian) weight sharing during the augmentation process.

- We design a hardware-aware neural architecture search strategy based on hybrid computing augmentation. We start training with an costly model and let some parts of it fade away to meet the hardware constraints in the training process. NAS will search for shrinking solutions to further boost accuracy.

## 2 RELATED WORKS

**Multiplication-Free NNs.** In order to reduce the intensive multiplication that occupies the main energy and time consumption, an important trend is to use hardware-friendly operators instead of multiplications. ShiftNet(Wu et al., 2018; Chen et al., 2019) believes that the Shift can be regarded as a special case of Depthwise Convolution, and proposes a zero-parameter, zero-flop convolution operator. DeepShift(Elhoushi et al., 2021) retains the calculation method of original convolution, but replaces the multiplication with bit-shift and bit-reversal. BNNs(Courbariaux et al., 2016; Lin et al., 2016; Rastegari et al., 2016) binarize the weight or activation to build DNNs consisting of sign changes, and get faster calculation in hardware by *xnor*. AdderNet (Chen et al., 2020; Song et al., 2021) chooses to replace multiplicative convolution with less expensive addition, and design an efficient hardware implementation(Wang et al., 2021). ShiftAddNet(You et al., 2020) combines bit-shift and add. It gets up to 196× energy savings on hardware as shown in Tab. 1. ShiftAddVit(You et al., 2023) puts this idea into vision transformer and performs hybrid computing through mixture of experts.

**Network Augmentation.** The tiny neural network is developing rapidly. Networks and optimization techniques designed for MCU have already appeared at present(Lin et al., 2020; 2021a). It is also possible to train under the 256KB memory limit(Lin et al., 2022). Due to the smaller capacity, the training of tiny NNs will have more challenges. Once-for-all(Cai et al., 2020) proposes the Progressive Shrinking training method, and finds that the accuracy of the obtained model is better than the same network that trained from scratch. Inspired by this result, NetAug(Cai et al., 2022) raises a point that tiny neural networks need more capacity rather than noise in training. Therefore, they chose a scheme that is the opposite of network structure regularization methods like Dropout(Srivastava et al., 2014), StochasticDepth(Huang et al., 2016), DropBlock(Ghiasi et al., 2018): expand the model width and let the large model lead the small model to achieve better accuracy through weight sharing.

Table 1: Hardware cost under 45nm CMOS.

| OPs | Format | Energy (pJ) |
|---|---|---|
| **Mult.** | FP32 | 3.7 |
| | FP16 | 0.9 |
| | INT32 | 3.1 |
| | INT8 | 0.2 |
| **Add** | FP32 | 1.1 |
| | FP16 | 0.4 |
| | INT32 | 0.1 |
| | INT8 | 0.03 |
| **Shift** | INT32 | 0.13 |
| | INT16 | 0.057 |
| | INT8 | 0.024 |

**Nerual Archtecture Search.** NAS has achieved amazing success in automating the design of efficient NN architectures(Liu et al., 2019b;a). In addition to obtain higher accuracy, some works include hardware performance of the model, such as latency(Tan et al., 2019; Wu et al., 2019) and memory(Lin et al., 2020), into the search. In parts that are closer to the hardware, NAS can also be used to explore faster operator implementations(Chen et al., 2018) and combine network structures for optimization(Lin et al., 2021b; Shi et al., 2022). BossNAS(Li et al., 2021) searched the network of hybrid CNN-transformers structure and ShiftAddNAS(You et al., 2022) for the first time constructed a search space with mixed multiplication and multiplication-free operators. However, unlike our target, networks the ShiftAddNAS focuses on are far beyond the hardware limitations of edge devices.

## 3 SHIFTADDAUG

In this section, we introduce the hybrid computing augmentation method and then present our heterogeneous weight sharing strategy to solve the weight-tearing problem. In the end, we introduce a new hardware-aware NAS method to get better multiplication-free tiny NNs.

### 3.1 PRELIMINARIES

**Shift.** For the shift operator, training is similar to the regular approach of linear or convolution operators with weight $W$, but round it to the nearest power of 2. During inference, use bit-shift and bit-reversal to efficiently get the same calculation result as Equ. 1. All inputs are quantized before calculation and dequantized when the output is obtained.

$$\begin{cases} \boldsymbol{S} = \texttt{sign}(\boldsymbol{W}) \\ \boldsymbol{P} = \texttt{round}(\log_2(|\boldsymbol{W}|)) \end{cases} \longrightarrow \begin{cases} \boldsymbol{Y} = \boldsymbol{X}\tilde{\boldsymbol{W}}_q^T = \boldsymbol{X}(\boldsymbol{S} \cdot 2^{\boldsymbol{P}})^T, & train. \\ \boldsymbol{Y} = \sum_{i,j}\sum_k \pm(X_{i,k} >> P_{k,j}), & infer. \end{cases} \tag{1}$$

**Add.** Add operator replaces multiplication in original convolutions with subtractions and $\ell_1$ distance, since subtractions can be easily reduced to additions by using complement code.

$$Y_{m,n,t} = -\sum_{i=0}^{d}\sum_{j=0}^{d}\sum_{k=0}^{c_{in}} |X_{m+i,n+j,k} - F_{i,j,k,t}|. \tag{2}$$

**NetAug.** Network Augmentation encourages the tiny NNs to work as a sub-model of a large model expanded in width. Based on shared weights, the target tiny NN and the augmented large model are jointly trained. The training loss and parameter updates are as follows:

$$\mathcal{L}_{aug} = \mathcal{L}(\boldsymbol{W}_t) + \alpha\mathcal{L}(\boldsymbol{W}_a), \quad \boldsymbol{W}_t^{n+1} = \boldsymbol{W}_t^n - \eta(\frac{\partial\mathcal{L}(\boldsymbol{W}_t^n)}{\partial\boldsymbol{W}_t^n} + \alpha\frac{\partial\mathcal{L}(\boldsymbol{W}_a^n)}{\partial\boldsymbol{W}_t^n}). \tag{3}$$

where $\mathcal{L}$ is the loss function, $\boldsymbol{W}_t$ is the weight of the target tiny NN, $\boldsymbol{W}_a$ is the weight of the augmented NN, and $\boldsymbol{W}_t$ is a subset of $\boldsymbol{W}_a$.

### 3.2 HYBRID COMPUTING AUGMENT

Our method is designed for tiny multiplication-free CNNs, introducing multiplication in the training process, and using only multiplication-free operators during inference as possible to improve the speed and save more energy. We combine the convolution kernels of different operators on the channel dimension as shown in Fig. 1. The target model will use multiplication-free convolution (MFConv, ShiftConv(Elhoushi et al., 2021) or AddConv(Chen et al., 2020) can be chosen), then take multiplicative convolution (MConv, i.e. original Conv) as the augmented part.

Since NetAug widens the channel and increases the expanding ratio of the Inverted Block, the input of each convolution in the large model can be conceptually split into the target part $\boldsymbol{X}_t$ and the augmented part $\boldsymbol{X}_a$, so does the output $\boldsymbol{Y}_t, \boldsymbol{Y}_a$. In our work, $\boldsymbol{X}_t$ and $\boldsymbol{Y}_t$ mainly carry information of MFConv, while $\boldsymbol{X}_A$ and $\boldsymbol{Y}_A$ are obtained by original Conv.

Here we mainly discuss three types of operators commonly used to build augmented tiny NNs: Convolution (Conv), Depthwise Convolution (DWConv), and Fully Connected (FC) layer. The hybrid computing augmentation for DWConv is the most intuitive. We only need to split the input into $\boldsymbol{X}_t$ and $\boldsymbol{X}_a$, then use MFConv and MConv to calculate respectively and connect the obtained $\boldsymbol{Y}_t$ and $\boldsymbol{Y}_a$ in the channel dimension. For Conv, We use all input $\boldsymbol{X}$ to get $\boldsymbol{Y}_a$ through MConv. But to get $\boldsymbol{Y}_t$, we still need to split the input and calculate it separately, and finally add the results. Since the FC layer is only used as a classification head, its output does not require augmentation. We divide the input and use Linear and ShiftLinear to calculate respectively, and add the results. If bias is used, it will be preferentially bounded to multiplication-free operators.

$$DWConv : \begin{cases} \boldsymbol{Y}_t = \texttt{MFConv}(\boldsymbol{X}_t) \\ \boldsymbol{Y}_a = \texttt{MConv}(\boldsymbol{X}_a) \\ \boldsymbol{Y} = \texttt{cat}(\boldsymbol{Y}_t, \boldsymbol{Y}_a) \end{cases}, \quad FC : \begin{cases} \boldsymbol{Y}_t = \texttt{ShiftLinear}(\boldsymbol{X}_t) \\ \boldsymbol{Y}_a = \texttt{Linear}(\boldsymbol{X}) \\ \boldsymbol{Y} = \boldsymbol{Y}_t + \boldsymbol{Y}_a \end{cases},$$

$$Conv : \begin{cases} \boldsymbol{Y}_t = \texttt{MFConv}(\boldsymbol{X}_t) + \texttt{MConv}(\boldsymbol{X}_a) \\ \boldsymbol{Y}_a = \texttt{MConv}(\boldsymbol{X}) \\ \boldsymbol{Y} = \texttt{cat}(\boldsymbol{Y}_t, \boldsymbol{Y}_a) \end{cases} \tag{4}$$

### 3.3 HETEROGENEOUS WEIGHT SHARING

**Dilemma.** Weight sharing strategy is widely used in one-shot neural architecture search(Guo et al., 2020; Yu et al., 2020) and multi-task learning(Ruder, 2017). In Network Augmentation, to learn the most important information in the large network, the $\ell_1$ norm is calculated for the weight of each

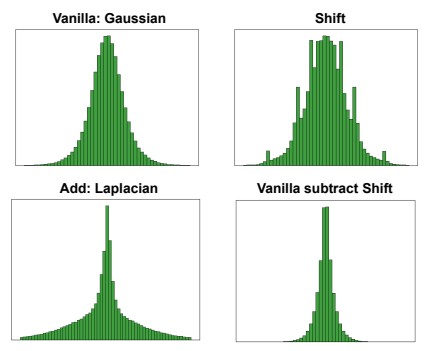 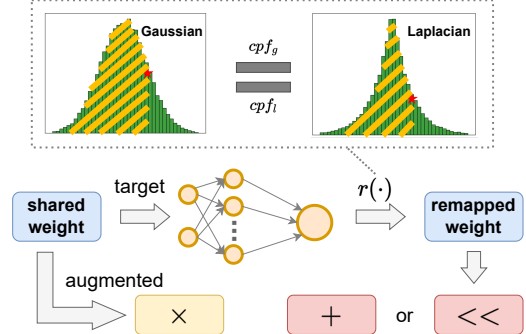

| (a). Weight distribution in different operators | (b). Remapping for heterogeneous weight sharing |

Figure 2: *Left*: Weight distribution of different convolution operators for MobileNetV2. Inconsistent weight distribution leads to tearing problems, making weight sharing difficult. *Right*: Through our weight remapping strategy, different operators can share a weight pool. For ShiftConv, the mapping result is only used as a bias.

channel at the end of every epoch. The important weights will be redirected to the target model. This looks like Channel-level Pruning(Mao et al., 2017), but different in the training process.

However, since the weight distribution of the multiplication-free operator is inconsistent with original Conv, it causes the weight tearing problem. As shown in Fig. 2(a), the weight in original Conv conform to Gaussian distribution, while ShiftConv has spikes at some special values. We find that the weight in ShiftConv is the one of original Conv plus a Laplace distribution with a small variance. The weight in AddConv conforms to the Laplace distribution. ShiftAddNas(You et al., 2022) adds a penalty term to the loss function, and guides the weight in heterogeneous operators to conform to the same distribution. Although this can alleviate the problem, it affects the network to achieve its maximum performance, which is more serious on tiny NNs with smaller capacity.

**Solution: heterogeneous weight sharing.** To solve the above dilemma, we propose a new heterogeneous weight sharing strategy for the shift and add operators. This method is based on original Conv and passes parameters to weights of different distribution types through a mapping function $\mathcal{R}(\cdot)$. Considering that the weights of original Conv and ShiftConv still have a deep relation, we hope to get a suitable bias to compensate for the weight when mapping to ShiftConv. For AddConv, we directly use the same method to get a new weight for replacement.

When mapping the Gaussian distribution to the Laplace distribution, we hope that the cumulative probability of the original value and mapping result are the same. Firstly, calculate the cumulative probability of the original weight in Gaussian. Then put the result in the percent point function of Laplacian. The workflow is shown in Fig. 2(b). The mean and standard deviation of the Gaussian can be calculated through the weights, but for the Laplace, these two values need to be determined through prior knowledge.

$$
\begin{aligned}
\boldsymbol{W}_l = \mathcal{R}(\boldsymbol{W}_g) = r(\texttt{FC}(\boldsymbol{W}_g)), \qquad & \texttt{cpf}_g(x) = \frac{1}{\sigma\sqrt{2\pi}}\int_{-\infty}^{x} e^{\left(-\frac{(x-\mu)^2}{2\sigma^2}\right)}\mathrm{d}x, \\
r(\cdot) = \texttt{ppf}_l(\texttt{cpf}_g(\cdot)), \qquad & \texttt{ppf}_l(x) = \mu - b * \texttt{sign}(x - \tfrac{1}{2}) * \ln(1 - 2\left|x - \tfrac{1}{2}\right|).
\end{aligned}
\tag{5}
$$

Where $\boldsymbol{W}_g$ is the weight in original Conv that conforms to the Gaussian distribution, and $\boldsymbol{W}_l$ is the weight obtained by mapping that conforms to the Laplace distribution. $\texttt{FC}$ is a fully connected layer. We need this because the weights don't fit the distribution perfectly. $\texttt{cpf}_g(\cdot)$ is the cumulative probability function of Gaussian, $\texttt{ppf}_l(\cdot)$ is the percentage point function of Laplace.

## 3.4 NERUAL ARCHTECTURE SEARCH

We take our proposed method one step further and use neural architecture search to design more efficient yet powerful multiplication-free models. Our method mainly aims to enhance the effect of hybrid computing augmentation.

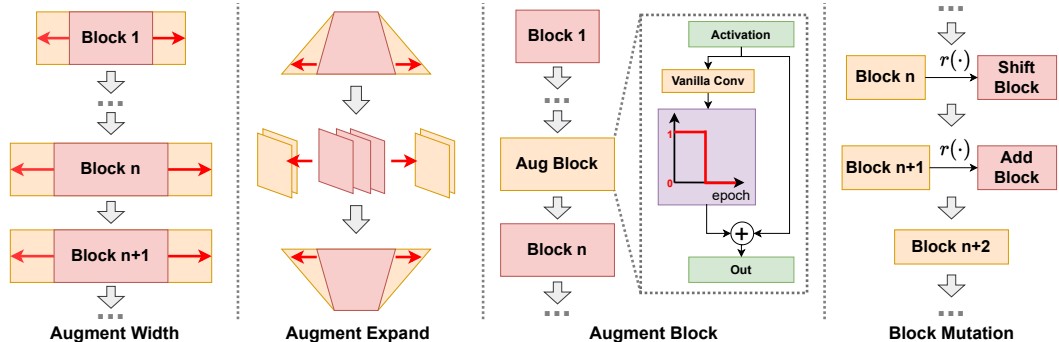

Figure 3: Methods used to construct search spaces. *Augment Width:* use `MConv` to widen the `MFConv` channel; *Augment Expand:* increase expand ratio of InvertedBlock, i.e. the channels of depthwise separable convolution; *Augment Block:* select some blocks and make them fade away during training for target model; *Block Mutation:* based on `MConv`, mutate the block into ShiftConv or AddConv.

We follow tinyNAS(Lin et al., 2020) to build our search space of model structure and set energy and latency limit of the target model to help us prioritize the elimination of some expensive model structures. Let the set of model structures that satisfy the hardware constraints be $\mathbb{T}$, and the set of all feasible model structures be $\mathbb{A}$. We first select the structure and operator type of the target model, and verify whether it belongs to the $\mathbb{T}$. If so, we create its counterpart, which is in the $\mathbb{A}$ but not in the $\mathbb{T}$, and transition it into the $\mathbb{T}$ during training. To achieve this, we use Augment Block and Block Mutation.

Table 2: Search space of ShiftAddAug.

| Block types | [Conv, Shift, Add] |
|---|---|
| width aug. multiples | [2.2, 2.4, 2.8, 3.2] |
| expand aug. multiples | [2.2, 2.4, 2.8, 3.2] |
| block aug. index | [None, 1, 2, 3] |

**Augment Block** is intended to insert some multiplication blocks into the backbone to help the target model extract more information during early training. It will then fade away in the target model but remain in the augmented one. In other words, this is an augmentation at the depth of the model. As for **Block Mutation**, although the operator type of the target model is determined at the beginning of each search, it lets the model use more powerful multiplicative operators in the early stage, and mutate to more efficient Shift or Add operators during the training process.

Combining the Width Augmentation and Expand Augmentation we used in section 3.2, we construct our search space according to Tab. 2. The schematic diagram of the four augmentation methods is shown in Fig. 3. We then perform an evolution search to find the best model within the search space.

## 4 EXPERIMENTS

### 4.1 SETUP

**Datasets.** We conduct experiments on several image classification datasets, including ImageNet-1K(Deng et al., 2009), CIFAR10(Krizhevsky, 2009), CIFAR100(Krizhevsky, 2009), Food101(Bossard et al., 2014), Flowers102 (Nilsback & Zisserman, 2008), Cars(Krause et al., 2013), Pets(Parkhi et al., 2012) and OpenEDS(Palmero et al., 2020) for segmentation task.

**Training Details.** We follow the training process in NetAug(Cai et al., 2022) and train models with batch size 128 using 2 GPUs. We use the SGD optimizer with Nesterov momentum 0.9 and weight decay 4e-5. By default, the Baseline and Shift models are trained for 250 epochs, and Add models are trained for 300 epochs. The initial learning rate is 0.05 and gradually decreases to 0 following the cosine schedule. Label smoothing is used with a factor of 0.1. Please refer to Appendix E for more details.

**Hardware Performance.** Since many works have verified the efficiency of shift and add on proprietary hardware(You et al., 2020; 2022; Wang et al., 2021; You et al., 2023), we follow their evaluation metrics. Hardware energy and latency are measured based on a simulator of Eyeriss-like

Table 3: ShiftAddAug vs. Multiplicative Baseline in terms of accuracy and efficiency on CIFAR100 classification tasks. ShiftAddAug not only improves the accuracy of popular tiny neural networks but also achieves better speed and energy efficiency. Please refer to Appendix.B for the specific meaning of each method.

| Model | Method | Params (M) | Mult (M) | Shift (M) | Add (M) | Accuracy(%) | Energy (mj) | Latency (ms) |
|---|---|---|---|---|---|---|---|---|
| MobileNetV2 w0.35 | Base / NetAug | 0.52 | 29.72 | 0 | 29.72 | 70.59 / 71.98 | 2.345 | 0.73 |
| | Shift / AugShift | 0.52 | 0 | 29.72 | 29.72 | 69.25 / **71.83** (↑2.58) | 0.74 | 0.246 |
| | Add / AugAdd | 0.52 | 4.52 | 0 | 56.88 | 67.85 / **69.38** (↑1.5) | 1.091 | 0.753 |
| MobileNetV3 w0.35 | Base / NetAug | 0.96 | 18.35 | 0 | 18.35 | 69.32 / 72.2 | 1.726 | 0.485 |
| | Shift / AugShift | 0.96 | 0 | 18.35 | 18.35 | 68.42 / **73.37** (↑4.95) | 0.536 | 0.16 |
| | Add / AugAdd | 0.96 | 3.5 | 0 | 34.34 | - / - | 0.699 | 0.512 |
| MCUNet | Base / NetAug | 0.59 | 65.72 | 0 | 65.72 | 71.38 / 73.15 | 4.28 | 1.682 |
| | Shift / AugShift | 0.59 | 0 | 65.72 | 65.72 | 70.87 / **74.59** (↑3.72) | 1.323 | 0.545 |
| | Add / AugAdd | 0.59 | 20.91 | 0 | 113.09 | 70.25 / **72.72** (↑2.47) | 2.345 | 1.72 |
| ProxylessNAS w0.35 | Base / NetAug | 0.63 | 34.56 | 0 | 34.56 | 70.86 / 72.32 | 2.471 | 0.883 |
| | Shift / AugShift | 0.63 | 0 | 34.56 | 34.56 | 70.54 / **73.86** (↑3.32) | 0.774 | 0.294 |
| | Add / AugAdd | 0.63 | 8.81 | 0 | 61.97 | 68.87 / **70.18** (↑1.31) | 1.281 | 0.881 |
| MobileNetV2 -Tiny | Base / NetAug | 0.35 | 27.31 | 0 | 27.31 | 69.3 / 71.62 | 2.161 | 0.67 |
| | Shift / AugShift | 0.35 | 0 | 27.31 | 27.31 | 68.29 / **71.89** (↑3.6) | 0.697 | 0.228 |
| | Add / AugAdd | 0.35 | 4.43 | 0 | 52.09 | 66.57 / **67.65** (↑1.08) | 0.999 | 0.693 |

Table 4: Accuracy of MobileNetV2 (w0.35) and MCUNet on more datasets. Training with ShiftAddAug can improve model performance without any overhead during inference on fine-grained classification tasks.

| Model | Methods | CIFAR10 | ImageNet | Food101 | Flower102 | Cars | Pets |
|---|---|---|---|---|---|---|---|
| MobileNetV2 - w0.35 | Shift | 88.59 | 51.92 | 72.99 | 92.25 | 72.83 | 75.4 |
| | AugShift | 92.51 | 53.86 | 74.67 | 96.08 | 74.47 | 79.59 |
| MCUNet | Shift | 90.61 | 56.45 | 78.46 | 95.59 | 80.51 | 79.67 |
| | AugShift | 93.08 | 57.34 | 79.96 | 97.06 | 83.29 | 83.95 |

hardware accelerator(Chen et al., 2017; Zhao et al., 2020), which calculates not only computational but also data movement energy.

## 4.2 SHIFTADDAUG VS. BASELINE

We validate our method on MobileNetV2, MobileNetV3, MCUNet, ProxylessNAS and MobileNetV2-Tiny. ShiftAddAug provides consistent accuracy improvements (average ↑2.82%) for ShiftConv augmentation over the multiplicative baselines. For AddConv augmentation, it improves the accuracy compared with direct training (average ↑1.59%). The resulting model will be faster (3.0× for Shift) and more energy-efficient (↓68.58% for Shift and ↓52.02% for Add) due to the use of hardware-friendly operators. As shown in Tab. 3, these multiplication-free operators usually hurt the performance of the network. Changing all operators to Shift will cause ↓0.82% accuracy drop on average compared to the multiplication baseline. But after using our method, the accuracy increased by ↑3.63% on average under the same energy cost. [1]

In addition, our method achieves higher results than multiplicative NetAug on some models (MobileNetV3:↑1.17%, MCUNet:↑1.44%, ProxylessNAS:↑1.54%). This means that our method enables the multiplication-free operator to be stronger than those of the original operator.

To verify the generality of our method, we also conduct experiments on more datasets. As shown in Tab. 4, our method can achieve ↑0.89% to ↑4.28% accuracy improvements on different datasets. Hybrid computing augmentation works better on smaller models and datasets with less classification. On Flower102, MobileNetV2-w0.35 has ↑3.83% accuracy improvements with our method, while MCUNet has only ↑1.47%. This shows that smaller model capacity can achieve better effect on this dataset. The larger the model, the smaller the gain brought by augmentation. The same phenomenon also occurs in CIFAR10. But for bigger datasets such as ImageNet, even if it is augmented, the

---

[1]Loss diverges when we use AddConv on MobileNetV3, both direct training and augmented training.

Table 5: ShiftAddAug vs. SOTA NAS method for hybrid operators in terms of accuracy and efficiency on CIFAR-10/100 classification tasks. ShiftAddAug can further improve the performance of the obtained multiplication-free model.

| Model | Method | Resolution | Mult (M) | Shift (M) | Add (M) | Accuracy(%) | MACs Saving |
|---|---|---|---|---|---|---|---|
| CIFAR10 | ShiftAddNas (Mult-free) | 32 | 2 | 26 | 38 | 91.32 | - |
| | ShiftAddNas | 32 | 17 | 19 | 58 | 95.83 | - |
| | **ShiftAddAug (Mult-free)** | 160 | 0.13 | 27 | 27.1 | **93.43**(↑2.11) | 17.9% |
| | **ShiftAddAug** | 96 | 12.3 | 14.5 | 28 | **95.92**(↑0.09) | 40.4% |
| CIFAR100 | ShiftAddNas (Mult-free) | 32 | 3 | 35 | 48 | 71.0 | - |
| | ShiftAddNas | 32 | 22 | 21 | 62 | 78.6 | - |
| | **ShiftAddAug (Mult-free)** | 160 | 0.13 | 33.5 | 33.6 | **74.61**(↑3.61) | 21.9% |
| | **ShiftAddAug** | 160 | 21 | 17 | 51 | 76.21 | 15.2% |
| | **ShiftAddAug** | 96 | 16.2 | 20.2 | 36.4 | **78.72**(↑0.12) | 33.8% |

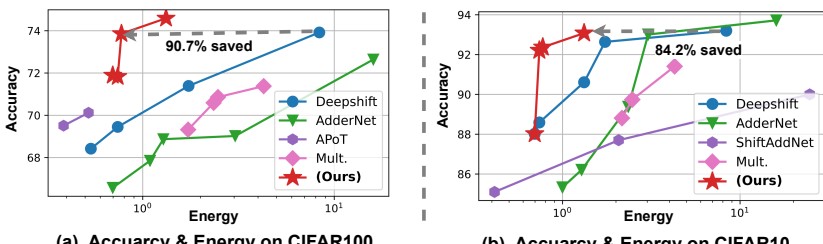

(a). Accuarcy & Energy on CIFAR100      (b). Accuarcy & Energy on CIFAR10

Figure 4: Accuracy and energy cost of ShiftAddAug over SOTA manually designed multiplication-free model and tiny multiplicative models. Tested on CIFAR-100/10.

capacity of the model is still not enough. It only achieves ↑1.94% for MobileNetV2-w0.35 and ↑0.89% for MCUNet on ImageNet. For segmentation task, please refer to Appendix. D.

### 4.3 SHIFTADDAUG VS. SOTA MULT.-FREE MODELS

We further compare ShiftAddAug over SOTA multiplication-free models, which are designed manually for tiny computing devices, on CIFAR-10/100 to evaluate its effectiveness. As shown in Fig. 4, the base models we use are smaller and have better energy performance. With ShiftAddAug, the accuracy still exceeds existing work. For DeepShift and AdderNet, our method boosts ↑0.67% and ↑1.95% accuracy on CIFAR100 with ↓84.17% and ↓91.7% energy saving. Compared with the SOTA shift quantization method APoT(Li et al., 2020), we achieve an improved accuracy of ↑3.8%. With the same accuracy on CIFAR10, our model saves ↓84.2% of the energy compared with Deepshift, and ↓56.45% of the energy compared with AdderNet.

### 4.4 SHIFTADDAUG WITH NEURAL ARCHITECTURE SEARCH

Based on hybrid computing augmentation, we introduce neural architecture search into our method to get stronger tiny neural networks. We conduct our experiments on CIFAR-10/100 and compare them with the results of ShiftAddNAS(You et al., 2022) under similar calculation amounts. As shown in Tab. 5, the multiplication-free model we obtained achieved higher accuracy (↑2.11% and ↑3.61%) than ShiftAddNas with FBNet(Wu et al., 2019) search space. For hybrid-computed models, we have to use a smaller input resolution (96 instead of 160) and larger models. While the input resolution of ShiftAddNas is 32, this gives us 9× the number of calculations at the same model size. Even so, we can still save 37.1% of calculations on average with similar accuracy.

### 4.5 ABLATION STUDY

**Hybrid Computing Augment.** In order to prove that hybrid computing works better, we add an experiment using only multiplication-free operators for augmentation. We exert experiments based on NetAug, and replace all the original operators with Shift operators. The difference from our method is that the Shift operator is also used in the augmentation part, while our method uses the multiplicative operator in it. As shown in Tab. 6, it yields an average accuracy improvement of ↑1.40%.

Table 6: The ablation study of hybrid computing augmentation and heterogeneous weight sharing in terms of accuracy on CIFAR100.

| Method | MobileNetV2 w0.35 | MobileNetV3 w0.35 | MCUNet | ProxylessNAS w0.35 | MobileNetV2 Tiny |
|---|---|---|---|---|---|
| Mult. baseline | 70.59 | 69.32 | 71.38 | 70.86 | 69.3 |
| To shift op | 69.25 | 68.42 | 70.87 | 70.54 | 68.29 |
| Aug. with Shift | 70.12 | 71.56 | 72.68 | 70.91 | 69.28 |
| Aug. with Hybrid Computation | 69.41 | 69.63 | 71.02 | 70.6 | 68.45 |
| **Aug. with HWS** | **71.83** | **73.37** | **74.59** | **73.86** | **71.89** |

Table 7: The ablation study of block augmentation and block mutation in terms of accuracy on CIFAR100. Results are obtained by neural architecture search.

| Method | Mult (M) | Shift (M) | Add (M) | Accuracy(%) | Energy (mj) | Latency (ms) |
|---|---|---|---|---|---|---|
| Aug. Width & Expand | 3.7 | 61 | 65 | 75.13 | 1.52 | 0.57 |
| Aug. Width & Expand & Block | 1.5 | 56 | 64.3 | 75.63 | 1.42 | 0.66 |
| Aug. Width & Expand, Mutation | 0.6 | 58 | 67 | 75.92 | 1.4 | 0.67 |
| **All** | **0.1** | **71.9** | **72** | **76.35** | **1.632** | **0.554** |

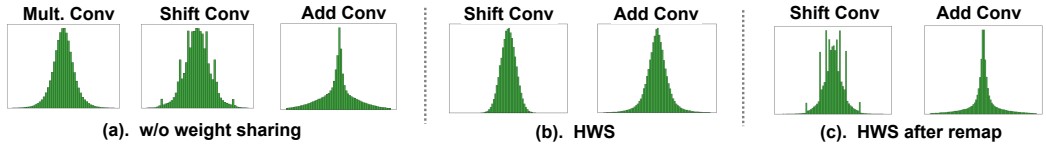

Figure 5: The weight distribution of original Conv / ShiftConv / AddConv layers

Then without using the heterogeneous weight sharing (HWS) method, only augmenting tiny NNs with multiplicative operator will cause ↓1.09% accuracy drop on average due to the weight tearing problem. However, the situation changed after we applied for HWS. Compared with using the shift operator for augmentation, the accuracy increased by ↑2.2%.

**Heterogeneous Weight Sharing.** Since the help of HWS on training results has been discussed above, here we visualize the weight distributions of Conv layers in tiny NNs under three scenarios, (a) w/o weight sharing; (b) heterogeneous weight sharing; (c) weight after remapped, as shown in Fig. 5. We consistently observe that the three operators exhibit different weight distributions without weight sharing. With our HWS, the saved weights are based on the original Conv and conform to Gaussian distribution. After remapping, the weights can show different distribution states in Shift/Add Conv calculations. Please refer to Appendix.C for more ablation study of HWS.

**Neural Architecture Search.** Our neural architecture search approach is dependent on our proposed hybrid compututing augmentation. And it can help the multiplication-free operator to be as strong as the original operator. Block augmentation and block mutation help us further improve the performance of multiplication-free tiny NNs. As shown in Tab. 7, under similar energy consumption and latency, block augmentation improves accuracy by ↑0.5%, and block mutation improves by ↑0.79%. Combining all method, the accuracy of the target model is increased by ↑1.22%.

## 5 CONCLUSION

In this paper, we propose ShiftAddAug for training multiplication-free tiny neural networks, which can greatly improve model accuracy without expanding the model size. It exceeds the multiplicative model in terms of accuracy under the same structure. It's achieved by putting the target multiplication-free tiny NN into a larger multiplicative NN to get auxiliary supervision. To relocate important weights into the target model, we also propose a novel heterogeneous weight sharing strategy to approach the tearing problem caused by inconsistent weight distribution. Based on the work above, we use neural architecture search to design more powerful models. Extensive experiments on image classification task consistently demonstrate the effectiveness of ShiftAddAug on the training process of multiplication-free tiny neural networks.

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

## A    ABLATION STUDY OF TRAINING SETTINGS

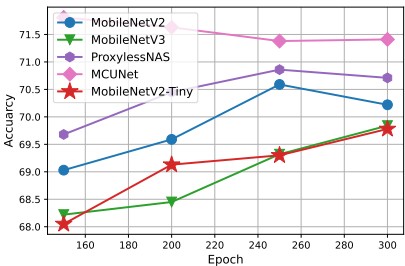

Figure 6: Baseline accuracy with training epochs.

More training epochs can improve accuracy as shown in Fig.6. However, training for too long may produce overfitting on the augmented model and require more training time. As a trade-off, we choose to train 250 epochs on the datasets such as CIFAR-10/100. A similar ablation study on ImageNet can be found in NetAug(Cai et al., 2022).

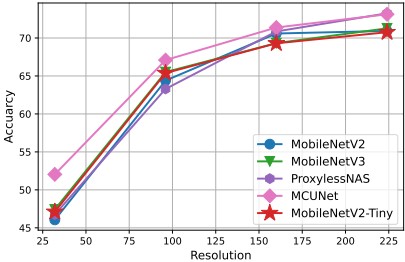

Figure 7: Baseline accuracy with different input resolution.

As shown in Fig. 6, higher resolution improves the accuracy but introduces more computation. We decided that all inputs should be resized to 160 during training and inference. Our model has a smaller capacity, allowing it to consume less energy than previous work with larger resolutions. This setting also means our experimental conclusions won't be limited to low-resolution datasets.

## B  THE SPECIFIC MEANING OF METHOD

- Base: directly trained multiplicative model.
- NetAug: multiplicative model with multiplicative augmentation.
- Shift: directly trained shift-model with ShiftConv in DeepShift(Elhoushi et al., 2021).
- AugShift: ShiftConv with multiplicative augmentation.
- Add: directly trained add-model with AddConv in AdderNet(Chen et al., 2020).
- AugAdd: AddConv with multiplicative augmentation.
- ShiftAddAug (Mult-free): results of neural architecture search with shift/add operator only
- ShiftAddAug: results of neural architecture search with shift/add/multi. operator

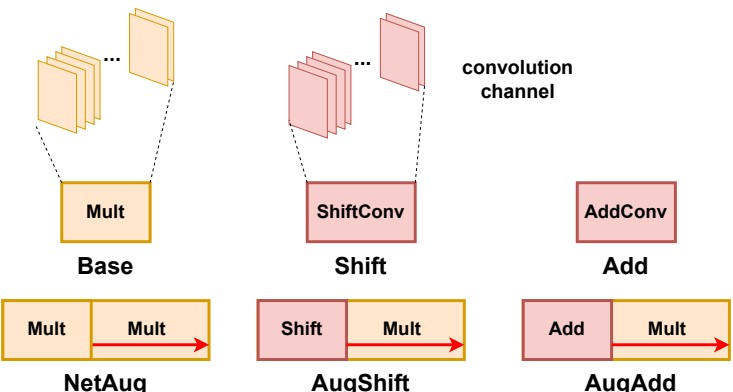

Figure 8: Schematic diagram of each method.

## C  MORE ABLATION STUDY OF HETEROGENEOUS WEIGHT SHARING

**The difference from ShiftAddNas.**  When we encountered the issue of weight tearing, we first thought of the solution in ShiftAddNAS(You et al., 2022). However, when we applied it, the training loss didn't converge, which made us believe that the method was not suitable for our training situation and it was difficult to compare performance. ShiftAddNas sorts the values of weights, dividing them into n groups from bottom to top, and then sets n learnable parameters to scale the weight values within each group. Our HWS strategy uses fully connected to remap the Conv kernel and use Equ.5 to handle different weight distributions. The obtained result is only added to the original weight as a bias, rather than applied directly. We use directly trained multiplicative and multiplication-free Conv weights as datasets to train the FC layer here, and freeze it in augmented training. We believe that our method has better training stability than ShiftAddNas. As shown in Tab.8, the ShiftAddNas method and direct mapping with learnable Linear will make it unable to train.

Table 8: The ablation study of different method for HWS

| Method | MobileNetV2 w0.35 | MobileNetV3 w0.35 | MCUNet | ProxylessNAS w0.35 | MobileNet-tiny |
|---|---|---|---|---|---|
| ShiftAddNAS | Nan | Nan | Nan | Nan | Nan |
| Linear remap | Nan | Nan | Nan | Nan | Nan |
| KL-loss only | 69.84 | 69.98 | 71.12 | 70.52 | 68.70 |
| Linear + skip connect + freeze | 71.02 | 72.70 | 74.44 | 72.99 | 71.08 |
| **Ours** | 71.83 | 73.37 | 74.59 | 73.86 | 71.89 |

**Is HWS a parameterization trick that can directly improve the target model?** We designed this ablation study to demonstrate that our HWS remapping method does not improve the accuracy of multiplication-free NNs by itself. On the contrary, it slightly damages the accuracy of the model. This kind of remapping is only a compensation method for different weight distributions, and will not produce gain for directly trained multiplication-free NNs.

Table 9: HWS is not a parameterization trick that can directly improve target model

| Method | MobileNetV2 w0.35 | MobileNetV3 w0.35 | MCUNet | ProxylessNAS w0.35 | MobileNet-tiny |
|---|---|---|---|---|---|
| w/o. augmentation, w/o. HWS | 69.25 | 68.42 | 70.87 | 70.54 | 68.29 |
| **w/o. augmentation, with HWS** | **68.32** | **68.10** | **71.13** | **69.88** | **68.02** |
| with augmentation, with HWS | 71.83 | 73.37 | 74.59 | 73.86 | 71.89 |

## D  SHIFTADDAUG IN SPECIFIC TASKS FOR IOT DEVICES

To demonstrate the effectiveness of our method in specific applications, we apply ShiftAddAug to a semantic segmentation task for IoT devices.

The segmentation of iris, pupil and sclera plays an important role in eye tracking in VR devices. To cope with this task, it is highly cost-effective to use fast and energy-efficient multiplication-free neural networks on such devices. OpenEDS(Palmero et al., 2020) is a large scale dataset of eye-images captured using a virtual-reality (VR) devices. We train each model from scratch for 100 epochs with a learning rate of 0.001 and batch size of 8. The mIoU(%) of the segmentation results of each model are shown in Tab.10 .

Table 10: ShiftAddAug on OpenEDS dataset for semantic segmentation task

| Method | MobileNetV2 w0.35 | ProxylessNAS w0.35 |
|---|---|---|
| Base/NetAug | 88.68 / 92.41 | 86.27 / 92.84 |
| Shift/**AugShift** | 88.01 / 94.52 | 86.01 / 95.12 |
| Add/**AugAdd** | 83.94 / 91.20 | 82.01 / 90.45 |

From the results shown in Fig.9, we can see that the model trained with augmentation will have fewer abnormal segmentation areas.

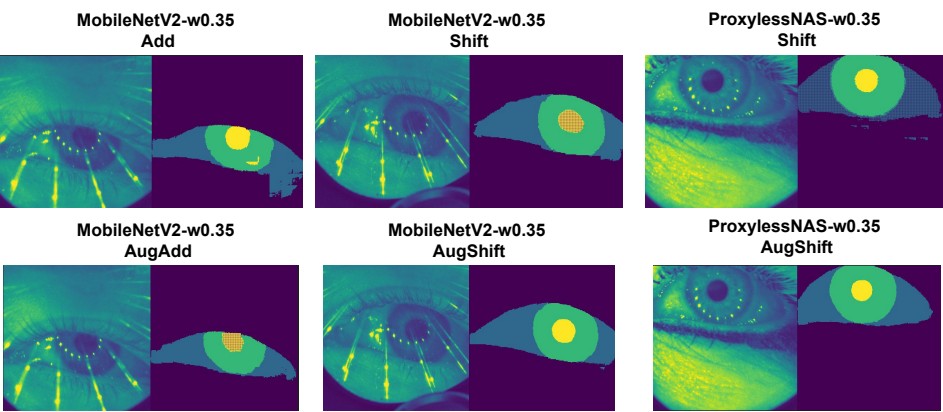

Figure 9: ShiftAddAug on OpenEDS.

## E  MORE TRAINING DETAILS

For ImageNet, we train models with batch size 768 on 6 GPUs. We use the SGD optimizer with Nesterov momentum of 0.9 and weight decay 4e-5. The initial learning rate is 0.15 and gradually decreases to 0 following the cosine schedule. Label smoothing is used with a factor of 0.1. We train 150 epochs with the multiplicative model and then finetune 100 epochs for the Shift model with the same setting. The models with AddConv are trained from scratch for 300 epochs.

For CIFAR10, we use pre-trained weights from CIFAR100 with original Conv and finetune 100 epochs for the Shift model. The models with AddConv are trained from scratch for 300 epochs on 2 GPUs. For other datasets, we load pre-trained weights from ImageNet and finetune with the same settings.

502 For the neural architecture search, we changed the model structures based on MCUNet and Mo-
503 bileNetV3 and pre-explored 100 model structures that met the hardware requirements. Energy con-
504 sumption and latency can be easily obtained. We start training from the model that meets the condi-
505 tions with the largest computational amount. Evolutionary algorithms are then used to explore other
506 model structures. Any setting that exceeds the hardware limit will be stopped early and output 0%
507 accuracy as a penalty. We trained them for 30 epochs for quick exploration and trained the top 10
508 for the full 300-epoch training.

509 For ShiftConv, its weights are quantized to 5 bits, and activations are quantized to 16 bits during
510 calculation. For AddConv, all calculations are performed under 32bit.

511 For HWS, we take the weights on the convolution kernel as input into the FC. The FC has a hidden
512 layer enlarged by a factor of 8. Then values goes through distribution remapping to get the output.
513 This section is pre-trained using independently trained model weights. We assume that this mapping
514 is generalizable and freezes its weights when training the final model.

## F  COMPARED WITH MULTIPLICATION-FREE NNs

| Method | Backbone | Resolution | Params(M) | Mult(M) | Shift(M) | Add(M) | CIFAR100 Accy(%) | Energy(mj) | Latency(ms) |
|---|---|---|---|---|---|---|---|---|---|
| DeepShift/ **AugShift** | MobileNetV2-w0.35 | 160 | 0.52 | 0 | 29.72 | 29.72 | 69.25 / **71.83** | 0.74 | 0.246 |
| | MobileNetV2-w1.0 | 32 | 2.4 | 0 | 94.72 | 94.72 | 72.39 | 1.749 | 0.821 |
| | MobileNetV2-Tiny | 160 | 0.35 | 0 | 27.31 | 27.31 | 68.29 / **71.89** | 0.697 | 0.228 |
| | MobileNetV3-w0.35 | 160 | 0.96 | 0 | 18.35 | 18.35 | 68.42 / **73.37** | 0.536 | 0.16 |
| | MCUNet | 160 | 0.59 | 0 | 65.72 | 65.72 | 70.87 / **74.59** | 1.323 | 0.545 |
| | ProxylessNAS-w0.35 | 160 | 0.63 | 0 | 34.56 | 34.56 | 70.54 / **73.86** | 0.774 | 0.294 |
| | ResNet-18 | 32 | 11.05 | 0 | 549.18 | 549.18 | 73.92 | 8.39 | 7.158 |
| | VGG19 | 32 | 20.08 | 0 | 399.21 | 399.21 | 62.68 | 7.603 | 5.192 |
| AdderNet/ **AugAdd** | MobileNetV2-w0.35 | 160 | 0.52 | 4.52 | 0 | 56.88 | 67.85 / **69.38** | 1.091 | 0.753 |
| | MobileNetV2-Tiny | 160 | 0.35 | 4.43 | 0 | 52.09 | 66.57 / **67.65** | 0.999 | 0.693 |
| | MCUNet | 160 | 0.59 | 20.91 | 0 | 113.09 | 70.25 / **72.72** | 2.345 | 1.72 |
| | ProxylessNAS-w0.35 | 160 | 0.63 | 8.81 | 0 | 61.97 | 68.87 / **70.18** | 1.281 | 0.881 |
| | ResNet-20 | 32 | 0.28 | 0.56 | 0 | 102.7 | 67.6 | 1.802 | 1.434 |
| | ResNet-32 | 32 | 0.47 | 0.56 | 0 | 174.6 | 69.02 | 3.038 | 2.428 |
| | VGG-small | 32 | 20.3 | 0 | 0 | 920 | 72.73 | 16 | 15.11 |
| ShiftAddNet | ResNet-20 | 32 | 1.13 | 0 | 40.8 | 127.17 | 58.5 | 2.07 | 2.315 |
| | VGG19 | 32 | 83.52 | 0 | 398 | 1380 | 65 | 24.68 | 25.632 |
| ShiftAddNAS | mult-free | 32 | - | 3 | 35 | 48 | 71.0 | - | - |
| | hybrid | 32 | - | 22 | 21 | 62 | 78.6 | - | - |
| **ShiftAddAug** | mult-free (NAS) | 160 | 1.3 | 0.13 | 33.5 | 33.6 | **74.61** | 0.851 | 0.264 |
| | hybrid (NAS) | 96 | 2.3 | 16.2 | 20.2 | 36.4 | **78.72** | 2.431 | 0.644 |

