# SUPPLEMENTARY MATERIAL OF SHIFTADDAUG: AUGMENT MULTIPLICATION-FREE TINY NEURAL NETWORK WITH HYBRID COMPUTATION

## 1 MORE RESULT ON CLASSIFICATION DATASETS

Table 1: Accuracy of MobileNetV3-w0.35 / ProxylessNAS-w0.35 / MobileNetV2-Tiny on more datasets.

| Model | Methods | CIFAR10 | ImageNet | Food101 | Flower102 |
|---|---|---|---|---|---|
| **MobileNetV3 - w0.35** | Shift | 88.85 | 54.19 | 73.3 | 93.82 |
| | **AugShift** | 92.83 | 56.07 | 75.1 | 96.28 |
| **ProxylessNAS - w0.35** | Shift | 87.71 | 55.4 | 75.66 | 94.31 |
| | **AugShift** | 92.37 | 56.3 | 77.02 | 96.76 |
| **MobileNetV2-Tiny** | Shift | 88.02 | 48.92 | 72.91 | 94.63 |
| | **AugShift** | 91.93 | 50.0 | 74.58 | 96.56 |

## 2 ABOUT KNOWLEDGE DISTILLATION

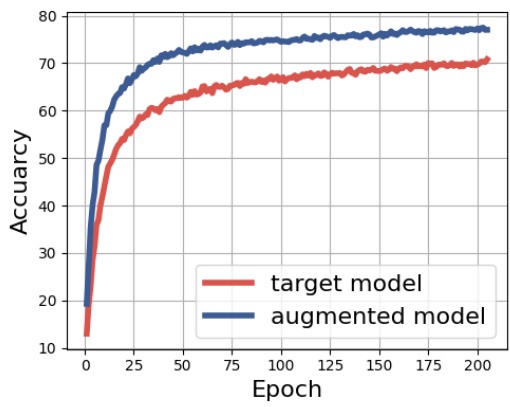

Figure 1: Training curve of MobileNetV2-w0.35 with ShiftAddAug.

As can be seen in Fig. 1, throughout the training process, the augmented model will have higher accuracy due to the larger capacity. It is a natural idea to use knowledge distillation to further improve the performance of the target model. Inplace Distillation(Yu & Huang, 2019) looks perfect for our situation. But in fact, it didn't work very well.

Inplace Distillation expects small models to gain more supervision from the soft labels of large models. It learns correct information while also learning biases in large models. Due to weight sharing, The large model and the small model in the same training step may exhibit similar biases.

Table 2: ShiftAddAug results on MobileNetV2 and MobileNetV3 using knowledge distillation.

| model | criterion | | | | origin |
|---|---|---|---|---|---|
| | KLLoss | | CELoss | | |
| | $\alpha = 0.9$ | $\alpha = 0.3$ | $\alpha = 0.9$ | $\alpha = 0.3$ | |
| MobileNetV2 - w0.35 | 65.7 | 68.93 | 64.89 | 69.02 | 71.83 |

This problem was not obvious in previous work. But multiplication-free operators are more unstable during training, making this problem serious in our case.

## 3 DISCUSSION ABOUT ADDCONV

In order to obtain a smaller model, using depthwise separable convolution with Inverted-Block(Sandler et al., 2018) is a must. But AdderNet's (Chen et al., 2020) implementation only works with ordinary convolutions. It will be slow and unstable in DWConv. So we keep DWConv as multiplication and convert the other parts to AddConv. But this still causes a loss of stability because the original AdderNet retains some multiplicative convolutions in the input and classification heads for higher accuracy. As you can see in Tab. 3, even though our method can boost accuracy compared with direct training, the result is still not ideal. This is a problem with AddConv itself.

Table 3: Accuracy of MobileNetV2-w0.35 / MCUNet with AddConv.

| Model | Methods | CIFAR10 | CIFAR100 | Food101 | Flower102 |
|---|---|---|---|---|---|
| **MobileNetV2 - w0.35** | Shift | 88.59 | 69.45 | 72.99 | 92.25 |
| | Add | 85.76 | 67.85 | 67.89 | 75.78 |
| | **AugAdd** | 87.21 | 69.38 | 68.62 | 78.92 |
| **MCUNet** | Shift | 90.61 | 70.87 | 78.46 | 95.59 |
| | Add | 89.38 | 70.25 | 70.6 | 78.63 |
| | **AugAdd** | 91.02 | 72.72 | 72.04 | 84.33 |

However, in the experiment of neural architecture search, keeping AddConv in the first few layers of the model helps improve accuracy. We keep the first 3 convolutions as AddConv instead of the original Conv, obtaining **0.43%** accuracy increase and some energy savings.