# OpenReview forum: "ShiftAddAug: Augment Multiplication-Free Tiny Neural Network with Hybrid Computation"
_ICLR.cc/2024/Conference — Submitted to ICLR 2024_

### Official Review · Reviewer_cMEX · 2023-10-29

**Soundness:** 3 good
**Presentation:** 4 excellent
**Contribution:** 4 excellent
**Rating:** 8
**Confidence:** 3

**Summary:**

This paper introduces "ShiftAddAug," a novel approach for training multiplication-free neural networks aiming to reduce energy costs. ShiftAddAug leverages costly multiplication operations during the training phase to enhance the activation of multiplication-free operations. These multiplication operations are subsequently deactivated during inference to avoid additional computational expenses. Additionally, the authors have developed a hardware-aware neural architecture search strategy rooted in a hybrid computing augmentation search space. This strategy dynamically reduces parts of the models to comply with specific hardware constraints throughout the training process. The results obtained from benchmarks such as CIFAR10/100 and ImageNet-1k, among others, demonstrate robust performance and significant gains in energy efficiency.

**Strengths:**

I appreciate the logical structure and clarity of this paper. The authors present their motivations compellingly, and the proposed ShiftAddAug method is both intuitive and seemingly effective, as evidenced by the strong results reported.

**Weaknesses:**

I have several questions that I hope the authors can clarify and expand upon to better understand the nuances of ShiftAddAug:

1. Could you elucidate how ShiftAddAug augments a baseline model? Specifically, in the context of convolutions, are additional channels created for multiplication-free (MF) operations on top of the existing ones? Or is there a division of existing channels between multiplicative and MF operations?

2. Regarding the dedicated input channels for shift/add operations, are they fixed throughout the training process? Figure 1 suggests the presence of a "gate" that directs input features, but this mechanism isn’t elaborated upon in the paper.

3. While the concept of heterogeneous weight sharing is intriguing, its practical application during training remains unclear. Are multiplication (M) weights dynamically mapped to MF weights, implying that operations aren't tied to specific input channels? If so, what determines the allocation of operations to particular channels?

4. After the neural architecture search (NAS) process, is further retraining of the resultant models necessary to achieve optimal performance? Does the performance reported in Table 5 directly stem from the NAS search, or is it the result of additional training?

5. Could the authors clarify the purpose of the last row in Table 3?

6. Regarding the behavior during inference, it's clear that multiplicative operations can be disabled, but can they also be entirely removed? The paper doesn't explicitly guarantee alignment between input/output channels among MF channels. Given this, are there concerns about potential computational waste due to this misalignment?

**Questions:**

see weaknesses

---

> ### Author Response · Authors · 2023-11-20
>
> We greatly appreciate your positive comments and constructive suggestions. Below are our detailed responses to your concerns.
>
> **W1: Could you elucidate how ShiftAddAug augments a baseline model?**
>
> Taking depthwise Conv as an example, let's assume we have 3 channels for MF Conv and 7 channels for original Conv. We will maintain a convolutional weight parameter with 10 channels in our code. We divide the weights of the first 3 channels as targets and calculate them using MF Conv. The weights of the remaining 7 channels are divided as the augmented part and calculated using the original  Conv. At the end of training, only the weights of the target part are exported.
>
> ---
>
> **W2: Regarding the dedicated input channels for shift/add operations, are they fixed throughout the training process? Figure 1 suggests the presence of a "gate" that directs input features, but this mechanism isn’t elaborated upon in the paper.**
>
> Yes, the channels used for shift/add operations are fixed. As mentioned above, we will manually maintain a set of parameters for each operator and only use the weights of the first few channels for shift/add operators. The rest will be used for augmented multiplicative operators. Therefore, it looks like a "gate" on each operator that controls the information flow.
>
> ---
>
> **W3: Are multiplication (M) weights dynamically mapped to MF weights, implying that operations aren't tied to specific input channels? If so, what determines the allocation of operations to particular channels?**
>
> Continuing from the explanation in *W1*, we keep using the weights of the first 3 channels for MF Conv, and the last 7 channels for augmented multiplicative Conv. At the end of each epoch, we will reorganize the weights of these 10 channels and move the important weights to the first 3 channels. We use the L1 norm to evaluate its importance. Since good weights in original Conv may not be good in MF Conv, we need HWS to cope with different weight distributions among different operators. When applying HWS, the weights of the first 3 channels are first mapped through Eqn. (5)，then being used for calculation. We will release the mapped result after calculation during training and only save it at the end of training for exportation.
>
> ---
>
> **W4: Is further retraining of the resultant models necessary to achieve optimal performance?**
>
> Yes, we do need additional training to obtain the final result. This setting is the same as MCUNet. Actually, This is necessary for our method, as superNet is multiplicative, and our NAS selects subNets on top of it for operator mutation and augmentation.
>
> ---
>
> **W5: Could the authors clarify the purpose of the last row in Table 3?**
>
> This is a serious editing error. Our intention was to place the best multiplication-free tiny NN we could achieve here. But considering that this data would duplicate the results in Tab. 5 and Tab. 6,  we abandoned this decision. Sorry for not deleting this line due to negligence. If you are interested, the data here should be:
>
> | Model            | Method | Params (M) | Mult (M) | Shift (M) | Add (M) | Accuracy(%) | Energy (mj) | Latency (ms) |
> | ---------------- | ------ | ---------- | -------- | --------- | ------- | ----------- | ----------- | ------------ |
> | ShiftAddAug-r160 | NAS    | 1.3        | 0.13     | 33.5      | 33.6    | 74.61       | 0.851       | 0.264        |
>
> ---
>
> **W6: Can multiplicative operations also be entirely removed?**
>
> Yes. In fact, we will first build a target model and then perform augmentation on the convolution channel. We guarantee alignment between input/output channels among MF channels.

---

### Official Review · Reviewer_J3s9 · 2023-10-30

**Soundness:** 2 fair
**Presentation:** 2 fair
**Contribution:** 2 fair
**Rating:** 3
**Confidence:** 4

**Summary:**

This work targets better accuracy vs. efficiency trade-offs for multiplication-free tiny neural networks. Specifically, it uses multiplication-based Conv in training to augment Shift-based Conv and Add-based Conv optimization for higher accuracy in the Shift/Add-based networks. The experiments on image classification tasks show the proposed ShiftAddAug framework can have higher accuracy (e.g., +4.05% on CIFAR-100) while reducing energy consumption (e.g., 68.9% reduction) as compared to multiplication-based networks.

**Strengths:**

1. Motivation: Current tiny DNNs are primarily designed using multiplication-based operators, often overlooking the more energy-efficient shift and add operators. Exploring shift/add-based tiny DNNs is, therefore, a worthwhile endeavor.

2. Comprehensive Review of Related Works and Preliminaries: The section on related works thoroughly covers existing multiplication-free networks. Additionally, the preliminaries provide a clear explanation of the shift and add operators utilized in this work.

3. Clear and Understandable Figures: The figures presented are clear, making the entire paper straightforward and easy to follow.

**Weaknesses:**

1. Quality of the Draft: It appears the authors may not have thoroughly proofread their draft before submission. In Table 3, the performance of the proposed ShiftAddAug is denoted as "xx". This is a crucial detail for comprehending the efficacy of the suggested framework.

2. Ambiguity in the Contribution of the Proposed NAS: From the details provided in Sec. 3.4, the introduced NAS, which is highlighted as the third contribution, seems to essentially apply tinyNAS (Lin et al., 2020) over the ShiftAddAug. The search space is presented in Table 2 without clarifying its design rationale. Consequently, the true value-add of the proposed NAS remains ambiguous.

3. Unclear Contribution of the Weight Sharing Strategy: The weight-tearing issue that the proposed weight sharing strategy addresses is previously identified in ShiftAddNAS (You et al., 2022). The weight mapping strategy delineated in Eq. 5 is similar to the approach in ShiftAddNAS, which employed a learnable transformation kernel, T (·), to transition shared weights from a Gaussian to a Laplacian distribution. However, there are no theoretical justifications or empirical findings illustrating why this strategy outperforms the one in ShiftAddNAS.

4. Concerns on the Accuracy of the Efficiency Metric: As mentioned in Sec. 4.1, the efficiency metric chosen for this study is the energy and latency reported by an Eyeriss-like hardware accelerator simulator. However, the referenced studies (Chen et al., 2017; Zhao et al., 2020) are designed for multiplication-based networks. The authors have overlooked elaborating on the specific modifications implemented to adapt the simulator for multiplication-free networks. Given this, it's debatable if the evaluation backdrop is fair for multiplication-based networks. A recommendation for the authors would be to utilize more reproducible metrics, such as the latency from the TVM-based Shift/Add execution in ShiftAddViT (You et al., 2023).

**Questions:**

Besides the previously listed weaknesses, I have the following questions:

1. The experiments exclusively consider the image classification task. How can it be asserted that this is the dominant task for IoT devices?

2. As indicated in Tab. 5, when introducing multiplication into the search space, the proposed ShiftAddAug displays a reduced accuracy compared to the baseline ShiftAddNAS (You et al., 2022). The given justification, which states "our method has given the multiplication-free operators strong capabilities, bridging the gap to the original operator", appears inconsistent. Notably, the ShiftAddNAS baseline actually boasts a similar count of Mult, Shift, and Add parameters. If the assertion were accurate, ShiftAddAug should outperform ShiftAddNAS in terms of accuracy. It seems more plausible that ShiftAddAug adversely affects the efficacy of multiplication-based operators. If this is the case, an exclusive emphasis on shift/add-only networks might be misplaced, as the existing multiplication hardware in IoT devices would remain underutilized.

---

> ### Author Response · Authors · 2023-11-20
>
> We greatly appreciate your positive comments and constructive suggestions. Below are our detailed responses to your concerns.
>
> **W1: Data in Tab.3 missing.**
>
> This is a serious editing error. Our intention was to place the best multiplication-free tiny NN we could achieve here. But this data would duplicate the results in Tab. 5 and Tab. 6. Sorry for not deleting this line due to negligence. The data here should be:
>
> | Model            | Method | Params (M) | Mult (M) | Shift (M) | Add (M) | Accuracy(%) | Energy (mj) | Latency (ms) |
> | - | - | - | - | - | - | - | - | - |
> | ShiftAddAug-r160 | NAS    | 1.3        | 0.13     | 33.5      | 33.6    | 74.61       | 0.851       | 0.264        |
>
> ---
>
> **W2:  The third contribution, seems to essentially apply tinyNAS over the ShiftAddAug.**
>
> **The difference from TinyNas.** TinyNAS finds the best multiplicative SubNet architecture from SuperNet. The SubNet selected here will directly meet the hardware requirements.  For our method, the SubNet should exceed the hardware limitation we set, and then gradually shrink into it during further retraining.
>
> **The design of search space.** In the program, we will first take out a SubNet1 that meets the hardware limitation. We use the "Block types" in Tab.2 to determine which operator to use for each layer. Then, we take out a deeper SubNet2 along SubNet1. The "block aug. index" in Tab.2 determines which layer will be shrunk during the training process. We start with multiplicative SubNet2 and use the "Augment Block" and "Block Mutation" in Fig.3 to change it to multiplication-free SubNet1 during training.
>
> "width aug. multiples" and "expand aug. multiples" come from NetAug and represent the expansion factor of the convolution channel and the number of channel expansion in MobilenetInvertedBlock.
>
> ---
>
> **W3: The weight mapping strategy delineated in Eq. 5 is similar to the approach in ShiftAddNAS.**
>
> **ShiftAddNas** sorts the values of weights, dividing them into n groups from bottom to top, and then sets n learnable parameters to scale the weight values within each group.  **Our HWS strategy** uses fully connected to remap the Conv kernel and use Equ(5) to handle different weight distributions.  The obtained result is only added to the original weight as a bias. We use directly trained multiplicative and multiplication-free Conv weights as datasets to train the FC layer here, and freeze it in augmented training. Our method has better training stability than ShiftAddNas. Please refer to **Appendix.C** for our updated ablation study.
>
> ---
>
> **W4: The Eyeriss-like hardware accelerators are designed for multiplication-based networks.**
>
> **About Eyeriss-like hardware accelerators.** We are very sorry that we do not have the ability to create hardware specifically for shift and add operators, so we use a simulator that takes the unit energy of computation, communication, data bits, and operator type into account. ShiftAddNas used Eyeriss-like hardware to evaluate energy and latency too, and the simulator does come from their [codebase](https://github.com/GATECH-EIC/ShiftAddNAS/blob/main/CV/retraining_hybrid/hw_utils.py).  Considering the outstanding contribution of this work, we believe that they have carefully considered the fair evaluation between different operators.
>
> **TVM-based Shift/Add execution.** Thank you very much for your suggestion. We use TVM to build the shift/add operator and deploy our model on RTX 3090.  We used batch_size = 8 for latency testing (ms).  The Baseline multiplicative model is evaluated in Pytorch. The input resolution is 160.
>
> |          | MobileNetV2 w0.35 | MobileNetV3 – w0.35 | MCUNet | ProxylessNAS -w0.35 | MobileNet-tiny |
> | - | - | - | - | - | - |
> | Baseline | 4.65              | 5.42                | 4.75   | 5.38                | 4.73           |
> | Shift    | 1.35              | 2.24                | 1.55   | 1.62                | 1.36           |
> | Add      | 3.76              | 4.93                | 4.68   | 5.19                | 3.864          |
>
> ---
>
> **Q1: Experiment on more task.**
>
> To demonstrate the effectiveness of our method in specific applications, we apply it to a semantic segmentation task. Please refer to **Appendix.D**.
>
> ---
>
> **Q2: Performance gap with ShiftAddNas**
>
> We need to clarify that "Mult (M), Shift (M), Add (M)" in Tab.5 means the number of calculations, not the count of Params. We used higher input resolution to evaluate the count of Mult, Shift, and Add calculations. Specifically, our input resolution is 160, while ShiftAddNas' is 32. This gives us **25$\times$** the count of calculations at the same model size.  We maintain this setting so that our hardware performance can be generalized to other high-resolution datasets.
>
> When we choose to use a larger model with a smaller input resolution (96 instead of 160), we can still beat the results of ShiftAddNas, saving 37.1% of calculations on average with similar accuracy. Please check our updated **Tab.5**

---

> ### Author Response · Authors · 2023-11-23
> **We Sincerely Look Forward to Your Post Rebuttal Feedback!**
>
> Dear Reviewer J3s9,
>
> We are following up to check whether our rebuttal responses have addressed your comments/concerns, and would be appreciative if you could let us know your feedback, thanks and have a good day!
>
> Best Regards.

---

### Official Review · Reviewer_x9yN · 2023-10-31

**Soundness:** 2 fair
**Presentation:** 1 poor
**Contribution:** 1 poor
**Rating:** 3
**Confidence:** 5

**Summary:**

The paper proposes to augment shiftadd operation kernel based training with traditional multiplication kernel based training to improve the performance of CNNs while doing inference with only shiftadd ops.

**Strengths:**

1. The idea of leveraging shiftAdd operation to improve compute bottleneck of CNNs is a useful and effective direction.

2. The paper is written well, apart from few sentences, example: the last sentence of related work (on NAS)

**Weaknesses:**

1. The paper's contribution needs improvement. The current draft is heavily based on ShiftAddNet and ShiftAddNAS.

2. In the abstract the authors compared the energy performance with traditional DNN, however, it should have been ShiftAddNet, if there is any. As it is already understandable due to the earlier publications in this line that shift-add ops based computation would incur energy saving over MAC based computation.

3. The idea of augmenting the training shiftadd kernel with multiplicative kernel would incur additional training compute and storage overhead, thus essentially altering the training recipe of the baseline shiftadd methods. Additionally, on device training and fine-tuning is a largely growing field, which is basically demeaned by this style of training compared to that of shiftaddnet.

4. Interestingly shiftAddNAS can be assumed as a superset of this work, which not only proposes the option of multiple compute kernel types, but also searches over them based on resource budget. Thus, I find it very hard to appreciate the current work in its current format.

5. The results are not comprehensive and the comparison baselines are not proper. The paper should be compared with ShiftAddNAS, ShiftAddNet, AdderNet, NetAug etc. Though having more comparisons are good, however, I am not sure why the authors compared with MCUNet, as there is already the next version in that family published > 1 year back,  MCUNet v2.

6. Results on ImageNet are incomprehensive.

7. Table 3 last row is not filled in!

**Questions:**

Please see weaknesses.

---

> ### Author Response · Authors · 2023-11-20
>
> We greatly appreciate your positive comments and constructive suggestions. Below are our detailed responses to your concerns.
>
> **W1/W4: The current draft is heavily based on ShiftAddNet and ShiftAddNAS. ShiftAddNAS can be assumed as a superset of this work.**
>
> We believe that our work is a lateral extension of the existing multiplication-free neural network.
>
> **The difference from ShiftAddNet.** ShiftAddNet proposes a method to jointly train the shift and add operators, while our work does not target the training method of multiplication-free operator itself.
>
> **The difference from ShiftAddNAS.** Although both use HWS, our method is more stable in training, while the ShiftAddNas method makes the loss not converge in our augmentation scenario. In addition, we used a completely different search process based on resource budget too. ShiftAddNas uses hybrid operators to train SuperNet, and selects the best SubNet on top of it that meets the hardware requirements. Our method starts from multiplicative SuperNet, selects a SubNet that exceeds the hardware performance limit, shrinks and mutates it to a multiplication-free NN that meets the requirements in further training. Please refer to our updated **Tab.5 and Appendix C**.
>
> ---
>
> **W2: The baseline in abstract should be ShiftAddNet.**
>
> Your suggestion helps to make this comparison more rigorous. We will make modifications to the abstract.
>
> ---
>
> **W3: Augmentation incur additional training overhead. It's not usable for on device training.**
>
> Yes, you are right and we do not claim our method is also good for on-device training. Our goal is to deploy multiplication-free NNs without inference overhead. However, this does not mean that the model trained with our method can not be further trained on device. At the end of training, we will **only export the target model** and discard the augmented part. Any on-device training methods can now be applied to the target model without overhead.
>
> ---
>
> **W5: The results are not comprehensive and the comparison baselines are not proper.** **Why compare MCUNet instead of MCUNet V2.**
>
> We need to clarify that the shift and add data in Tab.3 represent the results of Deepshift and AdderNet on these models, and the multiplication-based NetAug is also compared in Tab.3. The data of ShiftAddNet and ShiftAddNas are also reported in Fig. 4 and Tab. 5. In order to make the results clearer, we have organized the data and updated it to **Appendix F**.
>
> Regarding MCUNetV2,  MCUNet [Codebase](https://github.com/mit-han-lab/mcunet) links to two papers (V1 and V2). Models available in this repository are: ['mcunet-in0', 'mcunet-in1', 'mcunet-in2', 'mcunet-in3', 'mcunet-in4', 'mbv2-w0.35', 'proxyless-w0.3', 'mcunet-vww0', 'mcunet-vww1', 'mcunet-vww2', 'person-det']. We follow NetAug and use "mcunet-in3". In short, **the author of MCUNet did not distinguish the models of V1 and V2 in detail**, so we did not distinguish them in our paper too.
>
> ---
>
> **W6:  Results on ImageNet are incomprehensive.**
>
> We put the experimental results of the additional three models on Imagenet in the supplementary material. Due to the poor training efficiency of the AddConv operator and limited computing resources, it is difficult for us to use ImageNet as our dataset baseline. We are very sorry for this.
>
> |          | MobileNetV2 w0.35 | MobileNetV3 – w0.35 | MCUNet | ProxylessNAS -w0.35 | MobileNet-tiny |
> | - | - | - | - | - | - |
> | shift    | 51.92             | 54.19               | 56.45  | 55.4                | 48.92          |
> | augshift | 53.86             | 56.07               | 57.34  | 56.3                | 50.0           |
>
> ---
>
> **W7: Table 3 last row is not filled in!**
>
> This is a serious editing error. Our intention was to place the best multiplication-free tiny NN we could achieve here. But considering that this data would duplicate the results in Tab. 5 and Tab. 6,  we abandoned this decision. Sorry for not deleting this line due to negligence. If you are interested, the data here should be:
>
> | Model            | Method | Params (M) | Mult (M) | Shift (M) | Add (M) | Accuracy(%) | Energy (mj) | Latency (ms) |
> | ---------------- | ------ | ---------- | -------- | --------- | ------- | ----------- | ----------- | ------------ |
> | ShiftAddAug-r160 | NAS    | 1.3        | 0.13     | 33.5      | 33.6    | 74.61       | 0.851       | 0.264        |

---

> ### Author Response · Authors · 2023-11-23
> **We Sincerely Look Forward to Your Post Rebuttal Feedback!**
>
> Dear Reviewer x9yN,
>
> We are following up to check whether our rebuttal responses have addressed your comments/concerns, and would be appreciative if you could let us know your feedback, thanks and have a good day!
>
> Best Regards.

---

### Official Review · Reviewer_PuT1 · 2023-11-01

**Soundness:** 2 fair
**Presentation:** 2 fair
**Contribution:** 3 good
**Rating:** 6
**Confidence:** 4

**Summary:**

This work proposed a network augmentation methods for muliplication-free (MF) convolutional neural networks (CNNs). The augmented part is multiplicative and only exists during training to "condition" the training of the multiplication-free part. During inference, the augmented part will be disgarded; hence the inference latency and energy efficiency will not be compromised.

A key technical contribution of this work is the heterogeneous weight sharing between the MF part and the augmented part. The intuition is the observation of the distribution shift of the trained weights in the two parts. The authors proposed to use the so called "heterogeneous weight sharing with remapping" that maps the original conv weights to those in the MF convs so that the remapped weights approximately follow a Laplacian distribution. This weight sharing technique is essential for the success of the proposed method.

The authors conducted a number of experiments to show the effectiveness of the proposed method.

**Strengths:**

+ The idea of conditioning MF network training via network augmentation is interesting. While this idea is not new as the authors discussed in the related work section, the authors identified a unique numerical issue encountered when applying the methodology to MF network training, that is the weight tearing issue --- the inconsistency of weight distribution between the MF and mulicative parts. Solving this issue brought significant boost in performance to the proposed method.

**Weaknesses:**

**Clarity**. As the most important part of this work, the elaboration on the heterogeneous weight sharing technique is not clear enough. I feel confused about several parts of the technique when I was reading the paper.
- Between which two parts are the weights shared, and how? The authors are not quite clear (mathematically and technically) about this. According to the authors description, I guess the augmented convs (multiplicative) contain the original weights. The weights are mapped to those in the ML convs using Eqn (5).
- If my understanding is correct above, does it mean the augmented conv has to be in the exactly same size as the MF conv? If so, how do the [2.2, 2.4, 2.8, 3.2] multiples in Tab. 2 work for the NAS part?
- Is it correct understanding that there are no actually weights stored for MF part during training; instead they are generated with mapping (5) instantly?
- What is the consideration of adding a learnable FC layer in (5)? From my understanding, there are analytical way that maps data points from a Gaussian dist. to Lap dist, like optimal transport?
- Are the weights rounded to powers of 2 for ShiftConv?

**Experiment design**.
- The plain multiplicaive augmentation seems only to hurt the MF part without the heterogeneous weight sharing technique. Another possibility is that the weight sharing technique plays a role as a special parameterization trick. What if we only apply the parameterization without augmentation? Will this improve the performance?
- The MF augmentation seems to work well itself. Is the weight sharing also applied for this baseline? If so, what if we relax it? The point here is to see how the performance improves just by increasing the width of a MF network.

**Overall writing quality**. While the overall flow of the paper is ok, there are writing issues here and there. A incomlete list of issues:
- The ShiftAddAug-NAS row in Table 3 is missing. The MobileNetV3 Add/AddAug accuracy data is also missing without clarification.
- The punctuations and capitalizations in many places are wrong.
- The names of the compared methods are confusing. I recommend the authors use a dedicated paragraph to clarify the naming of counterparts compared in the experiments.

**Questions:**

See the weaknesses part.

---

> ### Author Response · Authors · 2023-11-20
>
> We greatly appreciate your positive comments and constructive suggestions. Below are our detailed responses to your concerns.
>
> **W1: Between which two parts are the weights shared, and how?**
>
> Taking depthwise Conv as an example, let's assume we have 3 channels for MF Conv and 7 channels for original Conv. We will maintain a weight parameter with 10 channels in our code. We divide the weights of the first 3 channels as targets and calculate by MF Conv. The weights of the remaining 7 channels are divided as the augmented part and calculated by the original  Conv. In the end, only the weights of the target part are exported. At the end of each epoch, we reorder these 10 channels and move the important weights to the first 3 channels. Weights are shared in this process.
>
> **W2: Does it mean the augmented conv has to be in the exactly same size as the MF conv?**
>
> No, they can have different sizes. Taking the "width aug. multiples " = 2.2 as an example, assuming the target model has $n$ channels, we maintain a weight of $2.2n$ channels and divide the first $n$ channels for MF Conv, and the last $1.2n$ channels for original Conv.
>
> **W3: Are there no actually weights stored for MF part during training?**
>
> Yes. Continuing from the explanation in *W1*, when applying HWS, the weights of the first 3 channels are first mapped through Eqn. (5)， then being used for calculation. We will release the mapped result after calculation during training and only save it at the end of training for exportation.
>
> **W4: What is the consideration of adding a learnable FC layer in (5).**
>
> We were inspired by the method of ShiftAddNas and hope to learn a mapping. But we need to clarify that the FC layer in (5) is trained in advance and **frozen**. So it's not learnable in the process of training the augmented model. If we set FC=Linear(1, 1) and discard $r(\cdot)$,   It will be similar to a simple OT mapping with linear kernel. We just changed the linear mapping to a 2-layer fully connected mapping, expecting it to have a stronger effect.
>
> **W5: Are the weights rounded to powers of 2 for ShiftConv?**
>
> Yes, during ShiftConv calculation, the weights will be quantized to powers of 2, and the activation will be quantized to int16.
>
> ---
>
>
> **Q1: Does the weight sharing technique play a role as a special parameterization trick?**
>
> If we only apply the parameterization without augmentation, it slightly damages the accuracy of the model. It is only a compensation method for different weight distributions, and will not produce gain for directly trained multiplication-free NNs.
>
> |                               | MobileNetV2 w0.35 | MobileNetV3 – w0.35 | MCUNet    | ProxylessNAS -w0.35 | MobileNet-tiny |
> | ----------------------------- | ----------------- | ------------------- | --------- | ------------------- | -------------- |
> | w/o Aug. ,  w/o HWS    | 69.25             | 68.42               | 70.87     | 70.54               | 68.29          |
> | **w/o Aug. ,  w/ HWS** | 68.32         | 68.1            | 71.13 | 69.88           | 68.02      |
> | w/ Aug. ,  w/ HWS      | 71.83             | 73.37               | 74.59     | 73.86               | 71.89          |
>
>
>
> **Q2:  Is the weight sharing also applied for this baseline? What if we relax it?**
>
> Yes, weight sharing is also in MF augmentation. Without weight sharing, the augmented part will have difficulty exchanging information with the target part.
>
> |                         | MobileNetV2 w0.35 | MobileNetV3 – w0.35 | MCUNet    | ProxylessNAS -w0.35 | MobileNet-tiny |
> | - | - | ------------ | --------- | ------------------- | -------------- |
> | Direct trained          | 69.25             | 68.42               | 70.87     | 70.54               | 68.29          |
> | **w/o. weight sharing** | 69.42        | 69.66           | 71.09 | 70.53           | 68.40      |
> | w/. weight sharing      | 70.12             | 71.56               | 72.68     | 70.91               | 69.28          |
>
> ---
>
> **E1: The ShiftAddAug-NAS row in Table 3 is missing. The MobileNetV3 Add/AddAug accuracy data is also missing.**
>
> About ShiftAddAug-NAS row, it is a serious editing error. Our intention was to place the best multiplication-free tiny NN we could achieve here. But this data would duplicate the results in Tab. 5 and Tab. 6. Sorry for not deleting this line due to negligence. The data here should be:
>
> | Model            | Method | Params (M) | Mult (M) | Shift (M) | Add (M) | Accuracy(%) | Energy (mj) | Latency (ms) |
> | - | - | - | - | - | - | - | - | - |
> | ShiftAddAug-r160 | NAS    | 1.3        | 0.13     | 33.5      | 33.6    | 74.61       | 0.851       | 0.264        |
>
> The MobileNetV3 Add/AddAug accuracy data is missing because we found that when we applied the AdderNet method to MobileNetV3, both baseline and AddAug were unable to train, resulting in loss divergence. We believe it is due to the incompatibility between AddConv itself and MobileNetV3. We will make modifications and updates to reduce confusion.

---

### Meta-Review · Area_Chair_Utne · 2023-12-06

**Metareview:**

The paper proposed "ShiftAddAug," a novel approach for training multiplication-free neural network. It mainly addresses the numerical issue of weight tearing, showing notable performance boost, including improvements in energy efficiency and accuracy in image classification tasks.

However, clarity is a major concern; the elaboration on the heterogeneous weight sharing technique is not clear enough, leading to confusion about its application and effectiveness. There are ambiguities in the paper, such as the unclear contribution of the proposed NAS and the weight-sharing strategy. The experimental design and results are questioned, including the comparison baselines and the impact of the weight-sharing technique on performance. There are also issues with the quality of the draft, including missing data in tables and writing issues that affect the paper's readability.

Given these concerns, the paper is recommended for rejection.

**Justification For Why Not Higher Score:**

See above summary

**Justification For Why Not Lower Score:**

N/A

---

### Decision · Program_Chairs · 2024-01-16

Reject